# Enhancement of the Immunostimulatory Effect of Phosphodiester CpG Oligodeoxynucleotides by an Antiparallel Guanine-Quadruplex Structural Scaffold

**DOI:** 10.3390/biom11111617

**Published:** 2021-11-01

**Authors:** Fika Ayu Safitri, Anh Thi Tram Tu, Kazuaki Hoshi, Miwako Shobo, Dandan Zhao, Arief Budi Witarto, Sony Heru Sumarsono, Ernawati Arifin Giri-Rachman, Kaori Tsukakoshi, Kazunori Ikebukuro, Tomohiko Yamazaki

**Affiliations:** 1Doctoral Program in Biology, School of Life Sciences and Technology, Institut Teknologi Bandung (ITB), Bandung 40132, West Java, Indonesia; fikaayusafitri1904@gmail.com; 2Research Center for Functional Materials, National Institute for Materials Science, 1-2-1, Sengen, Tsukuba 305-0047, Japan; TU.ThiTramAnh@nims.go.jp (A.T.T.T.); k.hoshi.oe@juntendo.ac.jp (K.H.); SHOBO.Miwako@nims.go.jp (M.S.); ZHAO.Dandan@nims.go.jp (D.Z.); 3Division of Life Science, Graduate School of Life Science, Hokkaido University, Kita 10, Nishi 8, Kita-ku, Sapporo 060-0808, Japan; 4Department of Molecular Biology and Biochemistry, Faculty of Medicine, Indonesia Defense University, Bogor 16810, West Java, Indonesia; arief.witarto@idu.ac.id; 5Physiology, Developmental Biology and Biomedical Sciences Research Group, School of Life Sciences and Technology, ITB, Bandung 40132, West Java, Indonesia; sonyheru@sith.itb.ac.id (S.H.S.); erna@sith.itb.ac.id (E.A.G.-R.); 6Department of Biotechnology and Life Science, Tokyo University of Agriculture and Technology, 2-24-16, Naka-cho, Koganei 184-8588, Japan; k-tsuka@cc.tuat.ac.jp (K.T.); ikebu@cc.tuat.ac.jp (K.I.)

**Keywords:** guanine quadruplex, high-order structure, monomeric, immunostimulatory activity, adjuvant

## Abstract

Guanine-quadruplex-based CpG oligodeoxynucleotides (G4 CpG ODNs) have been developed as potent immunostimulatory agents with reduced sensitivity to nucleases. We designed new monomeric G4 ODNs with an antiparallel topology using antiparallel type duplex/G4 ODNs as robust scaffolds, and we characterized their topology and effects on cytokine secretion. Based on circular dichroism analysis and quantification of mRNA levels of immunostimulatory cytokines, it was found that monomeric antiparallel G4 CpG ODNs containing two CpG motifs in the first functional loop, named G2.0.0, could maintain antiparallel topology and generate a high level of immunostimulatory cytokines in RAW264 mouse macrophage-like cell lines. We also found that the flanking sequence in the CpG motif altered the immunostimulatory effects. Gc2c.0.0 and Ga2c.0.0 are monomeric antiparallel G4 CpG ODNs with one cytosine in the 3′ terminal and one cytosine/adenine in the 5′ terminal of CpG motifs that maintained the same resistance to degradation in serum as G2.0.0 and improved interleukin-6 production in RAW264 and bone marrow-derived macrophages. The immunostimulatory activity of antiparallel G4 CpG ODNs is superior to that of linear natural CpG ODNs. These results provide insights for the rational design of highly potent CpG ODNs using antiparallel G4 as a robust scaffold.

## 1. Introduction

Synthetic oligodeoxynucleotides containing CpG motifs (CpG ODNs) are short segments of DNA that represent a type of pathogen-associated molecular pattern that can stimulate immune cells expressing toll-like receptor 9 (TLR9), such as dendritic cells and macrophages, and activate the immune response [1,2]. The initial recognition of CpG ODNs by the leucine-rich repeat in the extracellular domain of TLR9 leads to the dimerization of TLR9 that initiates an immune response by recruiting the MyD88-IRAK adaptor, resulting in the activation of multiple transcription factors, including NF-κB, AP-1, and IRF-7 [1]. Currently, synthetic CpG ODNs have great promise as vaccine adjuvants [2], immunotherapy agents for allergy [3], and anti-cancer agents [4]. However, ODNs with a phosphodiester (PD) backbone are susceptible to nuclease degradation, thus limiting their clinical application. Phosphorothioate-modified CpG ODNs have been well-studied as an alternative to enhance resistance against nucleases and are already utilized as vaccine adjuvants for the hepatitis B vaccine (HEPLISAV-B). However, phosphorothioate modification is associated with undesirable side effects, such as prolonged coagulation time, acute toxicity, and non-specific binding to proteins, raising concerns regarding its safety [5,6,7,8].

Non-canonical guanine-quadruplex (G4) structures can increase the nuclease resistance of CpG ODNs with a PD backbone [9,10,11]. The hybrid type G4 CpG ODNs with CpG motif in the central loop showed high immunostimulatory effect with acquired high nuclease resistance and cellular uptake [9]. However, the G4 structure frequently changes depending on the surrounding environment, including the type and concentration of coordinating metal cations (potassium, sodium, etc.) and the sequence and length of the loop regions [12,13]. Modifying the CpG ODNs in the G4 structure poses a challenge for assembly because of such changes in the G4 structure. Since the topology of G4 CpG ODNs changes depending on the length and sequence introduced into the loop region, it is difficult to accurately design G4 CpG ODNs. We aim to enhance the nuclease resistance and cellular uptake using the G4 structure, and it is important to control the topology of G4 structures for future rational design. To overcome the challenge of structural changes in G4, we used a well-known unimolecular quadruplex as a scaffold for CpG ODNs. We employed an antiparallel duplex/G4 DNA aptamer, RE31, which is a thrombin aptamer consisting of a 31-mer ODN that forms a unimolecular quadruplex comprising a stack of two guanine quartets (hereafter referred to as G-tetrad) connected by two external loops and one central loop with a 6-mer duplex region, as our basic design. RE31 shows high thrombin-inhibitory activity against bovine thrombin [14], human thrombin, and rat thrombin [15] and has a high affinity for thrombin and prothrombin [16]. RE31 shows a typical antiparallel circular dichroism (CD) spectrum with two positive peaks around 295 nm and 245 nm and a negative peak at 265 nm. The three-dimensional structure of RE31 has been previously determined [17]. Therefore, we designed a new high-order structured CpG ODN by introducing a CpG motif in the loop of RE31 while maintaining the anti-parallel topology of RE31.

In this study, we developed new monomeric G4 ODNs with an anti-parallel topology using the structurally analyzed duplex/G4 DNA, RE31, as a robust scaffold. The objective of this study was to maintain antiparallel duplex/G4 ODN structure and its stability when CpG motif was introduced to increase immunostimulatory activity. This finding provides a basis for future rational design of G4 CpG ODNS for vaccine adjuvant applications.

## 2. Materials and Methods

### 2.1. G4 Formation of CpG ODNs

HPLC-grade ODNs with PD backbones (Table 1) were purchased from Eurofins Genomics (Tokyo, Japan). The ODNs were dissolved in sterile Milli-Q water to prepare a stock solution of approximately 100 µM. The concentration of ODNs was quantified using a NanoDrop 2000 Spectrophotometer (Thermo Fisher Scientific, Waltham, MA, USA). In G-quadruplex formation, the stocked ODNs were diluted in Dulbecco’s phosphate-buffered saline (D-PBS; DS Pharma Biomedical, Osaka, Japan), heated at 95 °C for 5 min, followed by slow cooling to 4 °C for 30 min, and then stored at 4 °C for further experiments.

### 2.2. CD Measurement

To measure the CD spectrum, 2 µM CpG ODNs were dispensed into a 1-cm path length quartz cuvette (model T-18-ES-10, 1 cm path length, JASCO) and analyzed using a JASCO J-1500 spectropolarimeter at a scan speed of 50 nm/min, response time of 2.0 s, bandwidth of 1.0 nm, resolution of 0.2 nm, and sensitivity of 100 millidegrees (mdeg). CD spectra were recorded as the average of five replications over the spectral range of 220–320 nm at 10 °C or 25 °C. The D-PBS buffer spectrum was measured under the same conditions and subtracted as the background from each spectrum.

The CD melting experiments were performed from 5 °C to 90 °C with a temperature gradient of 1 °C/min based on the measurement of absorbance at 295 nm, as previously reported [9]. The Tm value was calculated by determining the upper and lower baseline values [18]. Linear regression was performed using GraphPad Prism 8 version 8.2.0 for Windows (GraphPad Software, La Jolla, CA, USA).

### 2.3. Polyacrylamide Gel Electrophoresis (PAGE)

To analyze the structure of the ODNs, 2 µL of 0.5 µM ODNs were applied to a 12% polyacrylamide gel (PAGE; 1 mm thick; TEFCO; Tokyo, Japan). Electrophoresis was performed at 4 °C at a constant voltage (180 V) in 0.5 × Tris-borate-EDTA (TBE) buffer (0.089 M, pH 8.3–8.5; Takara Bio, Shiga, Japan) supplemented with 4 mM KCl. A 10-base pair (bp) DNA ladder (Thermo Fisher Scientific) was used as a marker. The DNA in the gel was stained with SYBR Gold Nucleic Acid Gel stain (Thermo Fisher Scientific). Linear ODNs were used as references.

### 2.4. Stability of ODNs in Serum

The stability of the ODNs in serum was analyzed by adding 36 µL of 20% fetal bovine serum (FBS) into a tube, followed by the addition of 4 µL of 10 µM ODNs. The tube was incubated at 37 °C for 1, 2, 4, and 24 h. After the incubation, 4 µL of 250 mM EDTA was added to stop the reaction, and the tube was heated for 2 min at 80 °C. The ODNs were stored at 4 °C until PAGE analysis. Serum-treated ODNs were separated using e-PAGEL gel (15%, ATTO, Tokyo, Japan) in 0.5 × TBE buffer containing 4 mM KCl at a constant current of 21 mA at 4 °C. Quantification of the undegraded ODNs was performed using the Image Studio Lite software (LI-COR Biotechnology, Lincoln, NE, USA) by determining the fluorescence intensity of the corresponding bands stained with SYBR Gold.

### 2.5. Cell Culture

The mouse macrophage-like RAW264 cell line (RCB0535) was purchased from RIKEN BioResource Center (Tsukuba, Japan) and maintained in minimum essential medium (MEM) (Gibco, Carlsbad, CA, USA) supplemented with 10% (*v*/*v*) heat-inactivated FBS (Sigma-Aldrich, St Louis, MO, USA) and 1% (*v*/*v*) non-essential amino acid solution (NEAA, 100X, Wako Pure Chemical Industries, Osaka, Japan). Mouse dendritic cells (DC2.4, Sigma-Aldrich) were cultured in RPMI 1640 medium, GlutaMAX supplement, HEPES (Gibco, Carlsbad, CA, USA) supplemented with 10% heat-inactivated FBS, and 1% (*v*/*v*) NEAA. The cells were maintained at 37 °C in a humidified incubator with 5% CO_2_.

### 2.6. Preparation of Mouse Bone Marrow-Derived Macrophage Cells (BMDMs)

Seven-week-old female C57BL/6 mice (Charles River Laboratories Japan, Yokohama, Japan) were housed in a pathogen-free environment. All protocols were approved by the Animal Care and Use Committee of the National Institute for Materials Science (64-2020-5). To collect the BM cells, the mice were euthanized by intraperitoneal injection of an overdose of medetomidine–midazolam–butorphanol. Primary cells were harvested from the tibiae and femora of the female C57BL/6 mice. Red blood cells were removed from the BM using lysing buffer containing 155 mM NH_4_Cl, 10 mM KHCO_3_, and 0.1 mM EDTA. To induce BMDMs, 5 × 10^5^ BM cells were plated in growth medium (RPMI 1640 medium supplemented with GlutaMAX supplement, HEPES, 10% heat-inactivated FBS, 1% penicillin–streptomycin (P/S) (100X, Thermo Fisher Scientific), and 1% (*v*/*v*) NEAA) for 4 h to remove adherent cells. The supernatant was centrifuged at 300× *g* for 5 min at 4 °C. The collected cells were cultured in growth medium supplemented with 55 µM 2-mercaptoethanol (Gibco, Carlsbad, CA, USA) and 100 ng/mL recombinant murine macrophage colony stimulating factor (Peprotech, Cranbury, NJ, USA) for 13 days.

### 2.7. Immunostimulatory Assay

After three or four passages, RAW264 cells or DC2.4 cells were suspended in media supplemented with 1% P/S and seeded into a 96-well plate at a density of 1 × 10^5^ cells/well (5.59 × 10^5^ cells/mL, 180 μL). BMDMs were seeded in a 96-well plate at a density of 0.56 × 10^5^ cells/well (3.15 × 10^5^ cells/mL, 180 μL). After incubation for 18 h, 20 μL of 40 μM ODNs was added to the culture medium to a final concentration of 2 μM. The cells were then incubated for 24 h, and the supernatants and cells were collected. The relative transcript levels of interleukin (IL)-6, IL-12, and interferon (IFN)-β were evaluated by reverse transcription/real-time quantitative polymerase chain reaction (RT/RQ-PCR), following previously reported protocols, and normalized to that of glyceraldehyde-3-phosphate dehydrogenase (GAPDH) as housekeeping gene [9].

The concentration of murine interleukin (mIL)-6 secreted into the culture medium was measured using the mouse IL-6 ELISA Ready-Set-Go kit (Thermo Fisher Scientific) according to the manufacturer’s instructions.

### 2.8. Statistical Analysis

The data were statistically evaluated via one-way analysis of variance, followed by Tukey’s multiple range test for comparison between test groups and Dunnett’s multiple comparison test for comparison with the control group. All statistical analyses were performed using GraphPad Prism version 8.2.0.

## 3. Results

### 3.1. Effect of CpG Motif Position in Anti-Parallel Duplex/G4 ODNs on G4 Formation and Immunostimulant Function

All CpG ODN sequences evaluated in this study are listed in Table 1. To evaluate the effect of the number and position of CpG motifs on topology and cytokine production, we designed a new monomeric G4 CpG ODN with a CpG motif added to each functional loop in the duplex/G-quadruplex-based aptamer RE31, a structurally analyzed thrombin-binding aptamer [15,16,17]. RE31 consists of a duplex region with six pairs of nucleotides and a quadruplex region with two G-tetrads and three functional loops, hereafter called G0.0.0 (Figure 1). Based on the 5′ to 3′ positions, the molecule is named G4 (G), followed by the number of CpG motifs (GTCGTT) in the first, second, and third functional loops. The CpG motif modification in the loop is labeled as 0, 1, or 2, according to the number of CpG motifs representing the separation between the loops.

To investigate whether the position of the CpG motif in different functional loops affects the topology, we first synthesized three modified G0.0.0s containing two CpG motifs in each functional loop: G2.0.0, with two CpG motifs in the first loop, G0.2.0, with two CpG motifs in the second or central loop, and G0.0.2, with two CpG motifs in the third loop. The CD spectra of the designed G4-based CpG ODNs are shown in Figure 2. The CD measurement of G0.0.0 at 25 °C (black line) showed two positive peaks at 247 nm and 294 nm, with one negative peak at 265 nm (Figure 2a), which is characteristic of an antiparallel topology. CD measurement of CpG ODNs to study the effect of functional loop modifications showed that two CpG motifs in the first (G2.0.0) or third (G0.0.2) loops did not affect the antiparallel topology, which is seen in G0.0.0. However, when two CpG motifs were introduced into the second loop (G0.2.0), a positive peak at 295 nm and a negative peak at 265 nm were observed with a stepwise shift to 282 nm and 260 nm, respectively (Figure 2c). We assumed that the formation of G0.2.0 was not stable at 25 °C; therefore, we attempted to observe the topology at a lower temperature (10 °C). The CD spectrum of G0.2.0 at 10 °C did not show any features characteristic of antiparallel topology, similar to the spectrum at 25 °C. These results indicated that the insertion of the CpG motif in the first and third loops preserved the antiparallel topology, whereas CpG ODNs with additional CpG motifs in the second loop failed to maintain the antiparallel topology.

Next, we evaluated the immunostimulatory activity of G4-based CpG ODNs by examining mRNA expression in mouse macrophage-like RAW264 cells, since the transcript levels of inflammatory cytokine and type I interferon induced by CpG ODNs show the same tendency as protein secretion levels of these factors [9,19,20]. As shown in Figure 3, the position of the CpG motifs influenced cytokine production. G4 CpG ODNs having two CpG motifs in the first (G2.0.0) or second (G0.2.0) loops showed high IL-6, IL-12, and IFN-β mRNA induction. Conversely, G4 CpG ODN with CpG motif in the third loop had a lower immunostimulatory effect. To investigate the effect of the number of CpG motifs in the first functional loop, we employed G1.0.0, which had only one CpG motif. As shown in Appendix A, increasing the number of CpG motifs in the first functional loop was responsible for cytokine enhancement. Furthermore, G2.0.0 showed a dose-dependent immunostimulatory effect on RAW264 cells (Appendix A).

These results suggest that the CpG motif in the first loop in antiparallel type duplex/G4 maintains an antiparallel topology and can also induce the highest cytokine mRNA levels. We further investigated the enhancement of the immunostimulatory effect using G2.0.0.

### 3.2. Influence of Nucleotide Connectors between G-Tract and CpG Loop on G4 Formation and Immunostimulatory Activity

To improve the immunostimulatory effect of antiparallel type G4 CpG ODNs, we designed a variant of G2.0.0 with additional nucleotide connectors between G-tetrad and the CpG loop. Gc2c.0.0, in which cytosine (C) was inserted into the 5′ and 3′ ends of the CpG motif, showed an antiparallel topology with two positive peaks at 245 nm and 293 nm and a negative peak at 262 nm in CD spectroscopy at 25 °C (Figure 4a). The other three variants, Ga2a.0.0, Ga2c.0.0, and Gc2a.0.0, showed a shift in the positive peak at 295 nm to lower wavelengths, and the negative peak at 265 nm disappeared. However, upon CD spectrum measurement at 10 °C, Ga2a.0.0 and Ga2c.0.0 showed the same CD spectrum as that of G0.0.0, with two positive peaks at around 245 nm and 295 nm and one negative peak around 265 nm. However, Gc2a.0.0 did not show a typical CD spectrum characteristic of antiparallel topology even at 10 °C. We assumed that the insertion of nucleotides between the G-tetrad and the loop reduces the stability of G4 formation; however, cytosine insertion in both the 3′ and 5′ termini of the CpG motif had no effect on the G4 structure.

The immunostimulatory effects of these variants were evaluated. As shown in Figure 4e–g, an additional connector between the G-tetrad and the CpG motif could induce cytokine at levels higher than those obtained using G2.0.0 containing only two CpG motifs. In particular, the two variants with cytosine at the 3′-end of the CpG motif, Ga2c.0.0 and Gc2c.0.0, showed high cytokine induction ability.

Next, we examined variants in which two bases were inserted between the G-tract and the CpG motif. Figure 5 shows the CD spectra and cytokine production of G4 CpG ODN variants with two-base insertions. We designed four variants: Gac2ac.0.0, Gca2ac.0.0, Gca2ca.0.0, and Gac2ca.0.0. Only Gac2ac.0.0 showed typical CD spectra characteristics of antiparallel topology when measured at 25 °C. In the CD spectra of the other three variants, the positive peak at 295 nm was shifted to around 280–285 nm, and the negative peak at 265 nm was shifted to around 260 nm. The shifted peaks of Gca2ac.0.0, Gca2ca.0.0, and Gac2ca.0.0 did not revert to the typical antiparallel topology wavelengths even when measured at 10 °C. Moreover, the addition of three bases between the G-tract and CpG motif destabilized the antiparallel topology (Appendix A). The immunostimulatory activity of the two-base inserted variant was compared with that of the linear CpG ODN with two CpG motifs, G2.0.0, and single-base inserted variants. All nucleotide-inserted modifications showed a higher immunostimulatory effect than the linear CpG ODN and G2.0.0. The one-base inserted variant induced a much stronger immune response than the two-base inserted variant. This indicated that the optimized spacer length between G-tracts and CpG motifs increased the immunostimulatory efficiency of antiparallel G4 CpG ODNs.

### 3.3. Electrophoresis Analysis of Anti-Parallel G4 CpG ODNs

Next, we analyzed the structure of G4 CpG ODNs by PAGE. As shown in Figure 6, we compared the mobility of our designed CpG ODNs with that of linear ODNs with the same nucleotide base length. All G4 CpG ODNs migrated faster than the linear ODNs in both the presence and absence of potassium ions in the annealing condition. These results suggest that G4 CpG ODN forms a stable, compact structure that is independent of the presence of potassium ions.

### 3.4. Serum and Thermal Stability of G4 CpG ODNs

To evaluate the stability of antiparallel G4 CpG ODNs against nucleases in serum, ODNs were incubated in 20% (*v*/*v*) FBS at 37 °C for 0 h, 1 h, 2 h, 4 h, and 24 h and then subjected to electrophoresis (Figure 7). The results showed that the linear 2 CpG ODN was 75% and 90% degraded after 1-h and 4-h incubation, respectively. All G4 CpG ODNs degraded more slowly than the linear 2 CpG ODNs in serum. This result indicates that stability of ODNs in serum can be acquired by forming a G4 structure. In addition, the length of the inserted CpG sequence in the first external loop did not affect serum stability up to four bases.

To perform the thermodynamics analysis of designed G4 CpG ODNs, we used the CD melting temperature analysis from the unfolded formation (5 °C) to the folded formation (90 °C) (Appendix A). The melting temperatures of CpG ODNs were determined from the CD melting curves at 295 nm, as shown in Table 2. The Tm value of G0.0.0 without CpG was approximately 42 °C, whereas the Tm value of the CpG motif-inserted G4 CpG ODN decreased to below 35 °C. This indicated that the insertion of long sequences in the loop region induced thermal denaturation of the antiparallel G4 structure.

### 3.5. Immune Responses of Mouse Dendritic Cells and BMDMs

We assessed the immunostimulatory potential of antiparallel G4 CpG ODNs in mouse dendritic cells and primary macrophages. In this investigation, we employed G2.0.0, Gc2c.0.0, and Ga2c.0.0, as well as the linear 2 CpG motif. As shown in Figure 8, Gc2c.0.0 and Ga2c.0.0, antiparallel type G4 CpG ODNs with one cytosine in the 3′ flanking the sequence of CpG motifs, showed high immunostimulatory cytokine induction in DC2.4 dendritic cells and BMDMs, similar to the results obtained in RAW264 cells. We also examined the IL-6 protein production level in RAW264 cells and BMDMs induced by G4 CpG ODNs (Figure 9). IL-6 protein levels induced by Gc2c.0.0 and Ga2c.0.0 were significantly higher than those induced by G2.0.0, whereas linear 2 CpG did not induce IL-6. These results confirmed the positive correlation between the transcription level and the secretion level of IL-6. The results also showed that antiparallel G4 CpG ODNs can activate primary immune cells, not only cell lines.

## 4. Discussion

Recently, G4 structures have been found to increase nuclease tolerance in serum and enhance the immunostimulatory activity of CpG ODNs both in vitro and in vivo [9,10]. The formation of G4 structures has three possible topologies: parallel, antiparallel, and hybrid. The topology of G-quadruplex depends on loop lengths and sequences [9] and the type and concentration of cations [21,22]. In our study, we developed an intramolecular antiparallel G4 CpG ODN by inserting CpG motifs into functional loops of G4 ODNs. In the construction of new G4 CpG ODNs, G4 forming an antiparallel topology was selected by CD analysis, followed by the measurement of immunostimulatory activity. The results showed that by inserting the CpG motif into the first external loop of antiparallel G4, G4 CpG ODNs that maintain an antiparallel structure and exhibit a high immunostimulatory function can be obtained. Furthermore, we found that the additional single nucleotide connector between the G-tetrad and CpG motif as a flanking sequence improved the induction of cytokines in immune cells, including primary immune cells differentiated from mouse BMs.

In this study, we used the thrombin-inhibiting DNA aptamer, RE31, as a scaffold for preparing antiparallel G4 CpG ODNs. The position of the CpG motif in the G4 structure influenced the conformation of G4 CpG ODNs. G4 CpG ODNs with CpG motifs in the first and third loops maintained the antiparallel topology, whereas G4 CpG ODNs with CpG motifs in the second loop changed the topology to a random structure. Krauss et al. (2016) reported that the first and third loops of RE31 have important interactions with thrombin, and the second loop plays a crucial role in G-quadruplex formation [17]. The second loop (TGT) is involved in the transition from a linear structure to a G-quadruplex structure by linking through a reverse Hoogsteen hydrogen bond network [17]. Thus, in RE31, the first and third loops can be considered insertion sites for the CpG motifs that maintain the antiparallel G4 structure. The position of the CpG motif in the G4 structure also influences the immunostimulatory activity of G4 CpG ODNs. CpG ODNs are taken up by immune cells by endocytosis and bind to TLR9 in the endosomes/lysosomes compartment, inducing cytokinesis. The production level of marker proteins for immune activity, such as IL-6, IL-12, and IFN-β, depends on the interaction between CpG ODNs and TLR9. As shown in Figure 1, there are four guanine linkages (G_22_–G_25_) on the 3′-terminal side of the third loop of RE31, suggesting that cytosine and guanine of the CpG motif inserted into the third loop may interact with these polyguanines. This may have resulted in weaker interaction of the CpG motif in the third loop with TLR9, resulting in lower cytokine induction. To clarify this, further structural analysis of G4 CpG ODNs by NMR and X-ray structure analysis, as well as analysis of the interaction between TLR9 and CpG ODNs by immunoprecipitation and surface plasmon analysis, will be necessary.

We previously reported that spaces between the CpG motifs and G-tracts increased the stimulation efficiency of hybrid G4 CpG ODNs in RAW264 cells [9]. Based on our previous results, we added nucleotides between the G-tetrad and CpG motifs to enhance cytokine expression and observed the stability of the structure. We modified the length of the first functional loop by adding connectors between the CpG motifs and G-tetrad as follows: one space with A or C at the first position (between the G-tetrad and CpG motif in the 5′ to 3′ direction) and second position (between the CpG motif and G-tetrad in the 5′ to 3′ direction). We prepared eight variants of the antiparallel G4 CpG ODN, G2.0.0, of which only Gc2c.0.0 and Gac2ac.0.0 maintained their antiparallel topology at 25 °C. Other sequences, such as Ga2a.0.0, Ga2c.0.0, Gca2ac.0.0, and Gca2ca.0.0, showed CD spectra characteristic of antiparallel topology when the temperature was decreased to 10 °C, whereas the remaining variants showed CD spectra characteristic of non-antiparallel structures even at low temperatures. We evaluated the immunostimulatory effects of these variants on immune cells. The variants that maintained the antiparallel G4 topology induced high levels of IL-6, IL-12, and IFN-β mRNA transcription in RAW cells. These results indicated that stable retention of the antiparallel G4 structure resulted in high immunostimulatory activity.

Here, we stimulated dendritic cells and macrophages with our monomeric G4 CpG ODNs to observe their ability to produce cytokines that lead to signal communication between immune and non-immune cells [23,24,25]. These cytokines are responsible for the catabolic process called autophagy [26]. In RAW264 cells, Ga2c.0.0 and Gc2c.0.0 induced the highest mRNA and protein expression of mIL-6 compared to other designs, as determined using RT/RQ-PCR and ELISA, respectively. We also successfully stimulated mouse macrophage primary cells with monomeric G4 CpG ODNs. Macrophages mediate interactions with dendritic cells [27]; therefore, we stimulated BMDMs with monomeric ODNs and found that these cells highly expressed cytokines both at the mRNA transcription of IL-6, IL-12, and IFN-β and immunostimulatory IL-6 production levels. The ability of Ga2c.0.0 and Gc2c.0.0 to stimulate both cell lines and primary cells suggests their potential for further application in vivo. Given the evidence, our new antiparallel G4 has a robust scaffold function to maintain antiparallel G4 CpG ODNs and increases the nucleases resistance and high cytokine level in primary and line cells. However, in vivo studies are needed to demonstrate its application as a vaccine adjuvant.

## 5. Conclusions

We developed and characterized antiparallel duplex/G4-based CpG ODNs, which allowed us to understand the roles of each functional loop in the structure. We found that the distance between the G-tetrad and CpG motif is important to impart flexibility to the CpG motif that is necessary to be recognized by TLR9; however, the longest loop could not ensure the highest cytokine expression. Furthermore, one spacer was sufficient to stimulate and enhance cytokine production in both cell lines and primary cells. Our study provides useful information on G4 CpG ODNs for further in vivo applications.

## Figures and Tables

**Figure 1 biomolecules-11-01617-f001:**
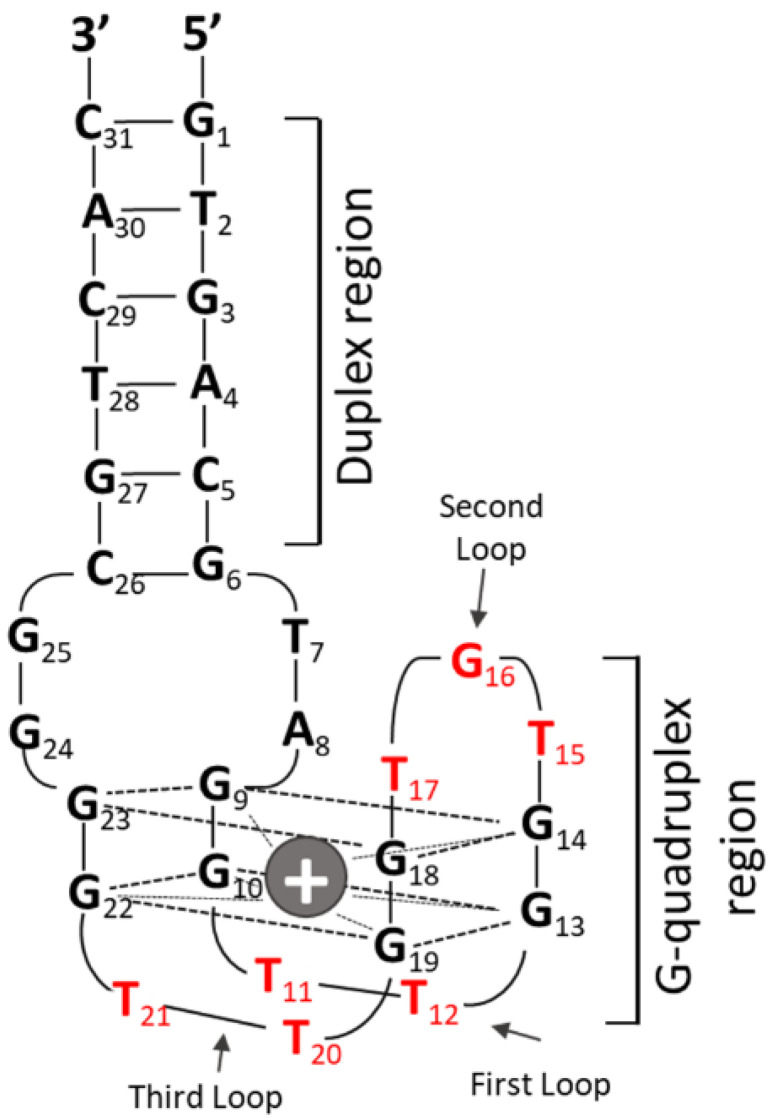
Structure of RE31 used as a scaffold to improve immunostimulatory function of CpG ODNs. The schematic represents the G0.0.0 design as the basic structure used in this study. The two G-tetrads in G0.0.0 are separated by three loops (L). Red colors in the G0.0.0 structures indicate loop sequences, where T_11_ and T_12_ form the first loop, T_15_, G_16_ and T_17_ form the second loop, and T_20_ and T_21_ form the third loop. The loops were modified by insertion of CpG motifs.

**Figure 2 biomolecules-11-01617-f002:**
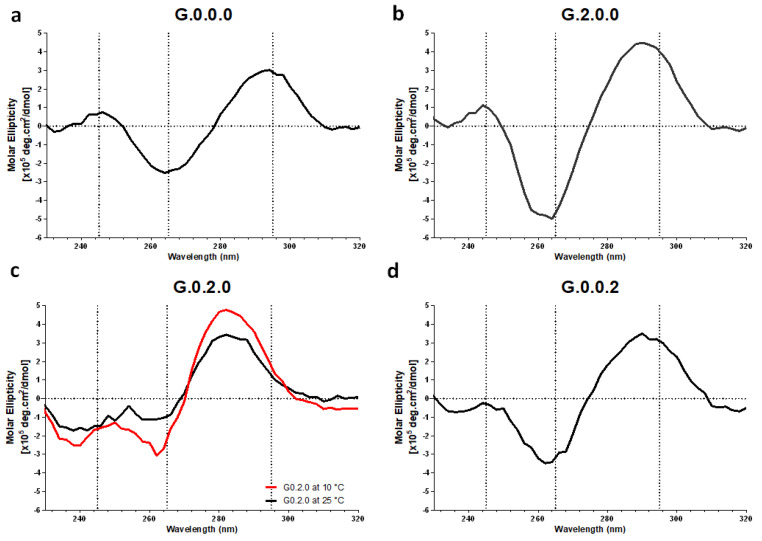
Circular dichroism spectra of (**a**) G0.0.0, (**b**) G2.0.0, (**c**) G0.2.0, and (**d**) G0.0.2 in D-PBS at 25 °C (black line) or 10 °C (red line). CpG ODNs concentration is 2 µM.

**Figure 3 biomolecules-11-01617-f003:**
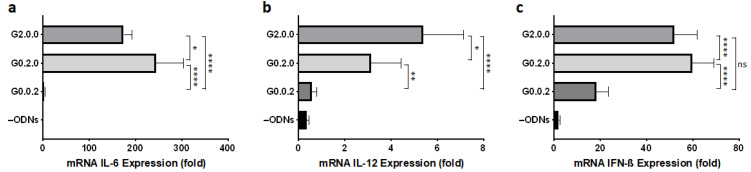
Induction of (**a**) IL-6, (**b**) IL-12, and (**c**) IFN-β mRNA transcription in RAW264 cells (1 × 10^5^ cells/well) stimulated with 2 µM duplex/G-quadruplex-based CpG ODNs. G2.0.0, G0.2.0, and G0.0.2 contain two CpG motifs in first, second, and third loop, respectively. –ODNs indicate RAW264 cells with only D-PBS buffer added. Data are presented as the mean ± SD (*n* = 5). **** *p* < 0.0001, ** *p* < 0.01, * *p* < 0.05, ns: not significantly different (one-way analysis of variance, Tukey’s multiple comparisons test for comparison with other groups).

**Figure 4 biomolecules-11-01617-f004:**
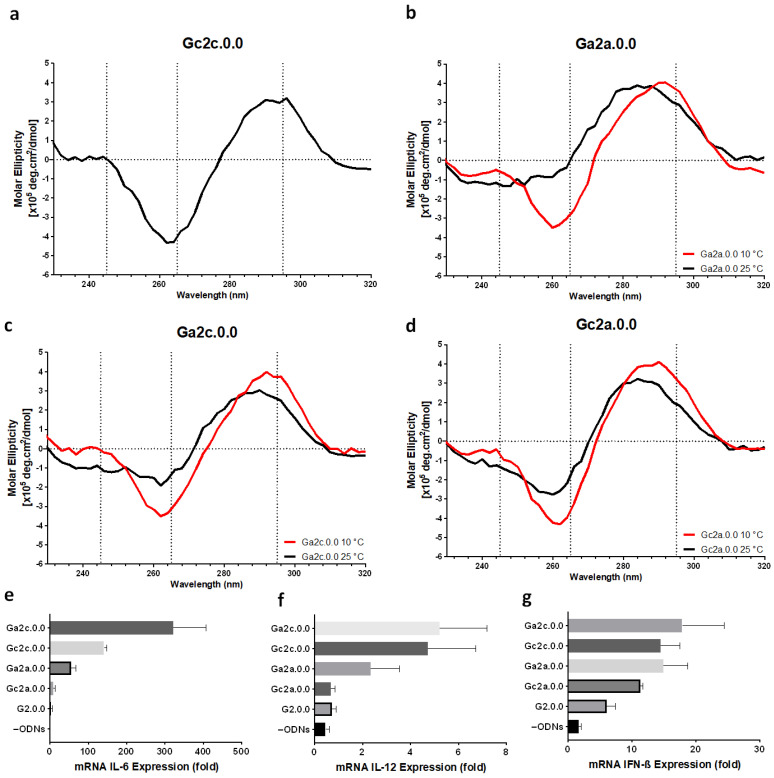
Effect of flanking sequence between the CpG motif and G-tetrad in the first loop on antiparallel G-quadruplex formation and cytokine induction. Circular dichroism spectra of (**a**) Gc2c.0.0, (**b**) Ga2a.0.0, (**c**) Ga2c.0.0, and (**d**) Gc2a.0.0 at 25 °C (black line) or 10 °C (red line) were observed. Relative expression level of (**e**) IL-6, (**f**) IL-12, and (**g**) IFN-ß mRNA transcription in RAW264 cells stimulated by G2.0.0, Ga2a.0.0, Gc2c.0.0, Ga2c.0.0, and Gc2a.0.0. –ODNs indicate RAW264 cells with only D-PBS buffer added. Data are presented as the mean ± SD (*n* = 5).

**Figure 5 biomolecules-11-01617-f005:**
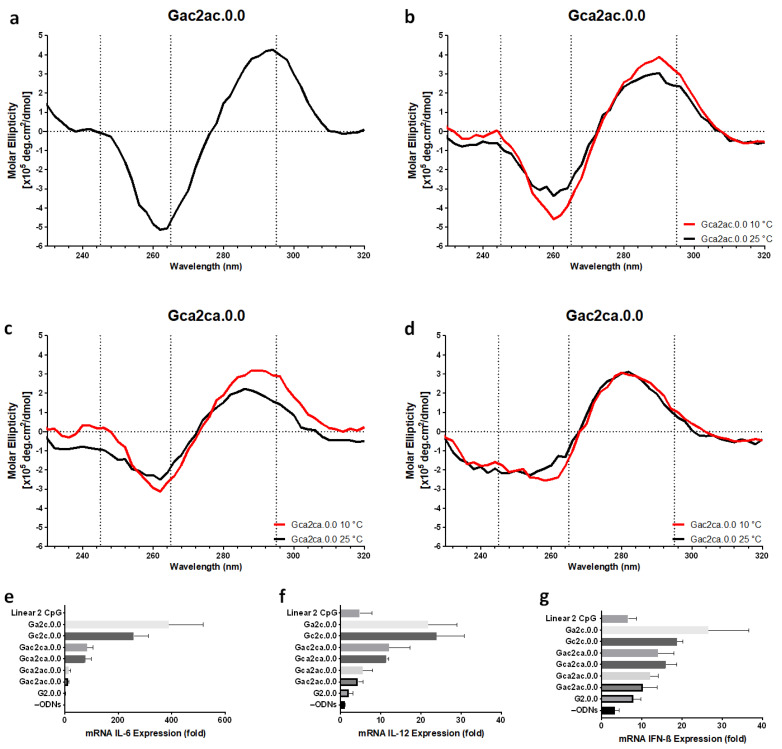
Effect of flanking sequence between the CpG motif and G-tetrad in the first loop on antiparallel G-quadruplex formation and cytokine induction. Circular dichroism spectra of (**a**) Gac2ac.0.0, (**b**) Gca2ac.0.0, (**c**) Gca2ca.0.0, and (**d**) Gac2ca.0.0 at 25 °C (black line) or 10 °C (red line) were observed. Relative expression level of (**e**) IL-6, (**f**) IL-12, and (**g**) IFN-ß mRNA transcription in RAW264 cells stimulated by Linear 2CpG, G2.0.0, Ga2a.0.0, Gc2c.0.0, Gac2ac.0.0, Gac2ca.0.0, Gca2ac.0.0, and Gca2ca.0.0. –ODNs indicate RAW264 cells with only D-PBS buffer added. Data are presented as the mean ± SD (*n* = 5).

**Figure 6 biomolecules-11-01617-f006:**
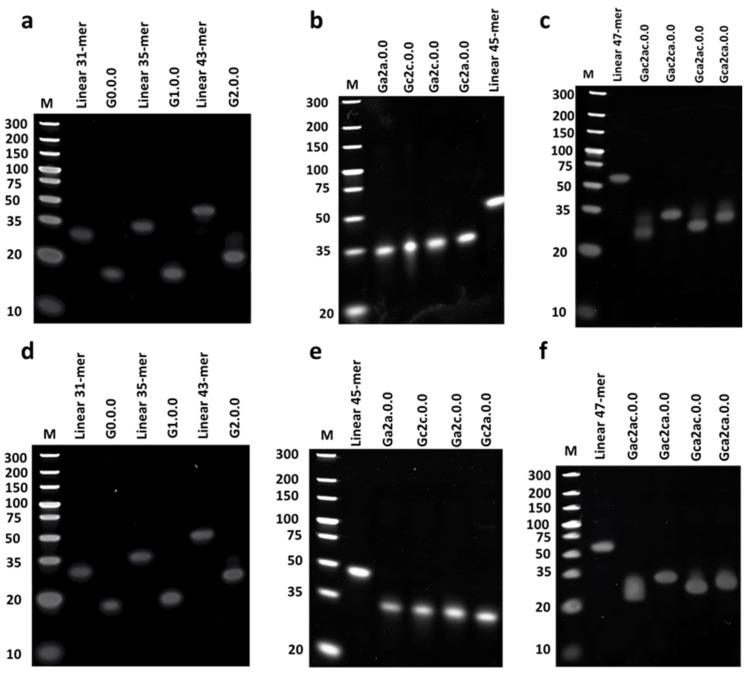
Polyacrylamide gel electrophoresis of G-quadruplex CpG ODNs. Samples were reconstructed in (**a**–**c**) D-PBS buffer (containing 4 mM K^+^ 150 mM Na^+^) and (**d**–**f**) Tris-HCl buffer by heating at 95 °C for 5 min followed by gradual cooling to 30 °C within 30 min, followed by cooling to 4 °C. Electrophoresis was performed using a 12% polyacrylamide gel in 0.5× Tris-Borate-EDTA buffer supplemented by 4 mM KCl. M is a 10-bp DNA marker.

**Figure 7 biomolecules-11-01617-f007:**
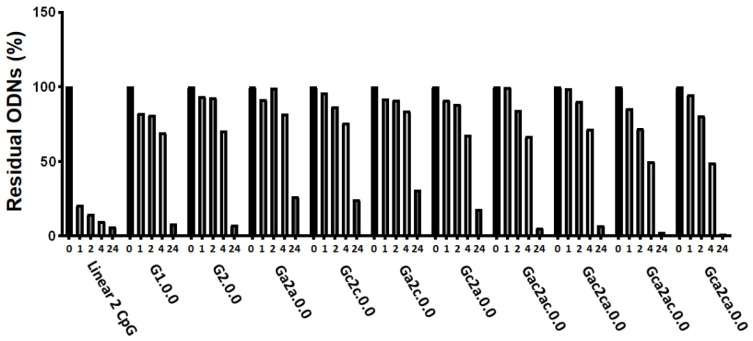
Degradation of G-quadruplex CpG ODNs in serum. Each ODN was incubated in 20% fetal bovine serum for 1, 2, 4, and 24 h at 37 °C. Percentage of residual ODN was analyzed by polyacrylamide gel electrophoresis with SYBR Gold staining.

**Figure 8 biomolecules-11-01617-f008:**
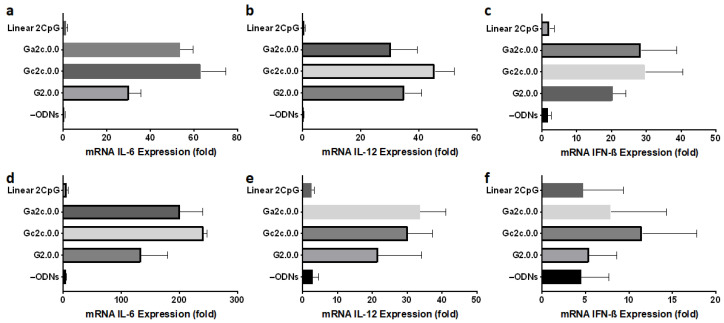
Immunostimulatory effect of antiparallel type CpG ODNs in (**a**–**c**) DC2.4 mouse dendritic cells and (**d**–**f**) bone marrow-derived macrophage cells (BMDMs). Messenger RNA expression levels of cytokines (**a**,**d**) IL-6, (**b**,**e**) IL-12, and (**c**,**f**) IFN-ß were determined using RT/RQ-PCR. The cells (DC2.4; 1 × 10^5^ cells/well, BMDMs; 0.56 × 10^5^ cells/well) were stimulated by 2 µM G2.0.0, Gc2c.0.0, and Ga2c.0.0 and linear 2CpG for 24 h. –ODNs indicate RAW264 cells with only D-PBS buffer added. Data are presented as mean ± SD (*n* = 5).

**Figure 9 biomolecules-11-01617-f009:**
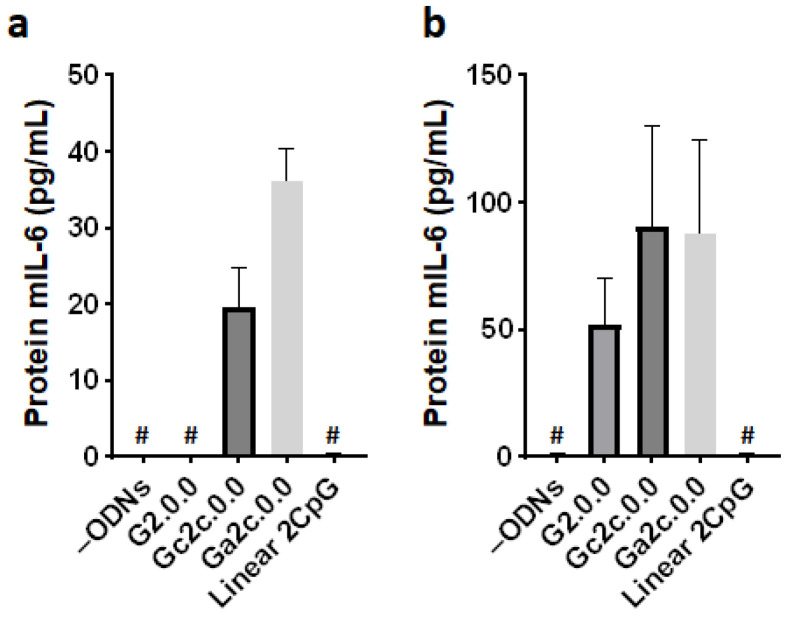
The level of IL-6 secreted into the culture medium induced by antiparallel type CpG ODNs. (**a**) RAW264 cells and (**b**) bone marrow-derived macrophage cells (BMDMs). The cells (RAW264; 1 × 10^5^ cells/well, BMDMs; 0.56 × 10^5^ cells/well) were stimulated by 2 µM G2.0.0, Gc2c.0.0, Ga2c.0.0, and linear 2CpG for 24 h, the supernatants were collected, and the secretion levels of interleukin-6 in medium were determined by ELISA. –ODNs indicate RAW264 cells with only D-PBS buffer added. Data are presented as mean ± SD (*n* = 5). The hash tag (#) indicates the lower detection limit (7.8 pg/mL).

**Table 1 biomolecules-11-01617-t001:** Sequences of loop modification oligonucleotides in this study.

Name	Sequence 5′ to 3′	Length
G0.0.0	GTGACGTAGGTTGGTGTGGTTGGGGCGTCAC	31
G2.0.0	GTGACGTAGGGTCGTTTTGTCGTTGGTGTGGTTGGGGCGTCAC	43
G0.2.0	GTGACGTAGGTTGGGTCGTTTTGTCGTTGGTTGGGGCGTCAC	42
G0.0.2	GTGACGTAGGTTGGTGTGGGTCGTTTTGTCGTTGGGGCGTCAC	43
G1.0.0	GTGACGTAGGGTCGTTGGTGTGGTTGGGGCGTCAC	35
Ga2a.0.0	GTGACGTAGG*A*GTCGTTTTGTCGTT*A*GGTGTGGTTGGGGCGTCAC	45
Gc2c.0.0	GTGACGTAGG*C*GTCGTTTTGTCGTT*C*GGTGTGGTTGGGGCGTCAC	45
Ga2c.0.0	GTGACGTAGG*A*GTCGTTTTGTCGTT*C*GGTGTGGTTGGGGCGTCAC	45
Gc2a.0.0	GTGACGTAGG*C*GTCGTTTTGTCGTT*A*GGTGTGGTTGGGGCGTCAC	45
Gac2ac.0.0	GTGACGTAGG*AC*GTCGTTTTGTCGTT*AC*GGTGTGGTTGGGGCGTCAC	47
Gac2ca.0.0	GTGACGTAGG*AC*GTCGTTTTGTCGTT*CA*GGTGTGGTTGGGGCGTCAC	47
Gca2ac.0.0	GTGACGTAGG*CA*GTCGTTTTGTCGTT*AC*GGTGTGGTTGGGGCGTCAC	47
Gca2ca.0.0	GTGACGTAGG*AC*GTCGTTTTGTCGTT*CA*GGTGTGGTTGGGGCGTCAC	47
Linear 2CpG	GTCGTTTTGTCGTT	

Note: CpG motifs are underlined. All ODNs have a phosphodiester backbone.

**Table 2 biomolecules-11-01617-t002:** Melting temperature at the first cooling and second heating of antiparallel ODNs with CpG motif substitutions in the first functional loop, with or without additional connectors, obtained from CD-melting curves in D-PBS.

Name	Loop Sequence			Tm at 1st Cooling (℃)	Tm at 2nd Heating (℃)
	1st	2nd	3rd		
G0.0.0	TT	TGT	TT	41.8	42.8
G1.0.0	GTCGTT	TGT	TT	40.2	41.2
G2.0.0	GTCGTTTTGTCGTT	TGT	TT	35.6	35.6
G0.0.2	TT	TGT	GTCGTTTTGTCGTT	32.9	33.4
Ga2a.0.0	AGTCGTTTTGTCGTTA	TGT	TT	33.6	34.9
Gc2c.0.0	CGTCGTTTTGTCGTTC	TGT	TT	32.5	32.8
Ga2c.0.0	AGTCGTTTTGTCGTTC	TGT	TT	30.2	29.5
Gc2a.0.0	CGTCGTTTTGTCGTTA	TGT	TT	25.9	25.2
Gac2ac.0.0	ACGTCGTTTTGTCGTTAC	TGT	TT	33.6	34.6
Gca2ac.0.0	CAGTCGTTTTGTCGTTAC	TGT	TT	32.5	34.7
Gca2ca.0.0	CAGTCGTTTTGTCGTTCA	TGT	TT	34.9	30.3

Note: CpG motifs are underlined. All ODNs have a phosphodiester backbone.

## Data Availability

Not applicable.

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
