# Peer review of "Enhancement of the Immunostimulatory Effect of Phosphodiester CpG Oligodeoxynucleotides by an Antiparallel Guanine-Quadruplex Structural Scaffold"

_biomolecules, 2021, doi:10.3390/biom11111617_

Round 1

Reviewer 1 Report

In this work, the authors developed a novel design of highly potent CpG ODNs using antiparallel G4 as a robust scaffold and primary demonstration of immunostimulatory effect. However, some revision should be done to improve the manuscript. It may be accepted if the authors address the following concerns:

1.To demonstrate the production of cytokines in immune experiment, we usually preform cytokine secretion detection in protein level using ELISA instead of qPCR detection in mRNA level, for the transcription of gene is not sure to translate in mammal cells. For the sake of reliability of immune response, ELISA is the gold standard to test cytokine production. So, figure3, figure4, figure 5 are suggested to use ELISA to give reliable demonstration on the immunostimulatory effect of different CpG oligodeoxynucleotides.

2.In the Abstract, the author writes” cytokine production in mouse macrophage cell lines, mouse dendritic cells, and bone marrow-derived primary cells”, but in the results figure 9, there is only one cytokine secretion detection-IL-6, in two types of cells-macrophage cell line and bone marrow 0derived macrophages. The lake of IL-12, IFN-β ELISA detection in DC2.4, RAW264.7 and BMDM should be explained. The IL-12, IFN-β ELISA detection in DC2.4, RAW264.7 and BMDM are suggested, and the IL-6 ELISA detection in DC2.4 are suggested.

  1. There is no evidence showing house keeping gene in the article, by which gene the target gene IL-6, IL12, IFN-β compared to? The house keeping gene such as GAPDH or β-actin should be mentioned.
  2. The abbreviation of bone marrow-derived macrophage cells is BMDM.
  3. The authors claims that the new version of CpG is resistant to nuclease degradation. Various nucleases such as DNase, Exo III, T5 exo should be tested.

Reviewer 2 Report

The manuscript is well written and organized. The use of CpG dinucleotides to induce immune response is coupled with antiparallel guanine-quadruplex structures based on the known RE31 structure with 2 antiparallel quadruplexes. The analyses of the structure based on a variety of sequence variants is reasonably thorough and the immune response of mouse dendritic cells and bone marrow macrophages well documented.

There are only a few, very minor, comments.

line 82: change the last sentence to "The 3D structure of RE31 has been previously determined."

In Table 1 and Table 2, for completeness add "CpG motifs are underlined."

line 132: change "indication" to "incubation".

Round 2

Reviewer 1 Report

The authors addressed the concerns.